# Characterization of the Activities of Vorinostat Against *Toxoplasma gondii*

**DOI:** 10.3390/ijms26020795

**Published:** 2025-01-18

**Authors:** Ting Zeng, Chun-Xue Zhou, Dai-Ang Liu, Xiao-Yan Zhao, Xu-Dian An, Zhi-Rong Liu, Hong-Nan Qu, Bing Han, Huai-Yu Zhou

**Affiliations:** 1Department of Pathogen Biology, School of Basic Medical Sciences, Cheeloo College of Medicine, Shandong University, Jinan 250012, China; 13974157491@163.com (T.Z.); ljwcmykycx@163.com (D.-A.L.); 15736031707@163.com (X.-Y.Z.); axd19990912@139.com (X.-D.A.); quhn@sdu.edu.cn (H.-N.Q.); bing.han@sdu.edu.cn (B.H.); 2Shandong Public Health Clinical Center, Cheeloo College of Medicine, Shandong University, Jinan 250012, China; liuzhirong@sdu.edu.cn

**Keywords:** *Toxoplasma gondii*, vorinostat, oxidative stress, mouse

## Abstract

*Toxoplasma gondii* is a globally widespread pathogen of significant veterinary and medical importance, causing abortion or congenital disease in humans and other warm-blooded animals. Nevertheless, the current treatment options are restricted and sometimes result in toxic side effects. Hence, it is essential to discover drugs that demonstrate potent anti-*Toxoplasma* activity. Herein, we found that vorinostat, a pan-HDAC inhibitor, exhibited an IC_50_ value of 260.1 nM against the *T. gondii* RH strain and a selectivity index (SI) > 800 with respect to HFF cells. Vorinostat disrupted the entire lytic cycle of *T. gondii* in vitro. Proteome analysis indicated that vorinostat remarkably perturbed the protein expression of *T. gondii*, and proteins involved in “DNA replication” and “membrane” were significantly dysregulated. Furthermore, we found that vorinostat significantly enhanced ROS production and induced parasite apoptosis. Importantly, vorinostat could prolong survival in a murine model. Our findings reveal that vorinostat is effective against *T. gondii* both in vitro and in vivo, suggesting its potential as a therapeutic option for human toxoplasmosis.

## 1. Introduction

*Toxoplasma gondii* is an opportunistic protozoon that is widely distributed throughout the world. Domestic cats and other felines serve as its specific definitive hosts, while all non-feline animals, including humans, are regarded as intermediate hosts. Approximately one-third of the world’s population is positive for *T. gondii* infection [1]. The main feature of *Toxoplasma* infection during the acute infection stage is the rapid proliferation of tachyzoites, which may lead to *Toxoplasma* encephalitis, multiple organ lesions, and even death. When the host has a strong immunity, *T. gondii* undergoes a transformation from tachyzoite to bradyzoite and enters the chronic infection stage, where bradyzoites reside within tissue cysts, mainly in muscle and brain cells [2]. People often acquire *Toxoplasma* infection by consuming undercooked meat containing cysts or by ingesting food or water contaminated with sporulated oocysts.

In recent years, evidence has accumulated indicating that chronic *T. gondii* infection is linked to some mental diseases, such as depression and schizophrenia [3]. It is particularly notable that when pregnant women are infected with *T. gondii*, the parasite can traverse the placental barrier to infect the fetus, leading to retinitis choroiditis, hydrocephalus, intellectual deficiency, and even stillbirth or abortion [4]. Additionally, *T. gondii* infections occur more frequently and severely among individuals with compromised immune systems. For instance, an elevated incidence of *Toxoplasma* encephalitis has been documented in countries with a high prevalence of HIV and low coverage of antiretroviral therapy [5]. Numerous epidemiological studies have affirmed that the infection rate of *T. gondii* in patients with malignant tumors is significantly higher than that in healthy individuals, mainly attributed to the weakened immune function of patients with malignant tumors, making them more prone to *Toxoplasma* infections [6,7,8]. Due to the prolonged use of immunosuppressive drugs, hematopoietic stem cell transplants and solid organ transplants have been associated with an augmented risk of contracting *T. gondii* infection [9]. Delays in diagnosing the infection after organ transplantation can have grave consequences, such as the sudden onset of shock, organ dysfunction, and even death [10]. Toxoplasmosis in individuals with compromised immune systems is most frequently the consequence of the reactivation of latent infection. The current therapy for toxoplasmosis depends on a few drugs, among which the combination of pyrimethamine and sulfadiazine constitutes the preferred therapeutic regimen. However, most of the currently available drugs mainly suppress the growth of tachyzoites and often have severe side effects [11]. Therefore, it is necessary to develop novel drugs for different types of toxoplasmosis.

Histone acetylation is one of the most extensively studied posttranslational modifications. The status of histone acetylation is regulated by histone acetyltransferases (HATs) and histone deacetylases (HDACs) [12]. When the level of histone deacetylation rises, the level of acetylation relatively declines, which can result in abnormal cell cycle and metabolic alterations, thereby inducing tumors and neurodegeneration [13]. HDAC inhibitors (HDACis) can exert anti-tumor effects by promoting cell differentiation, blocking the cell cycle, inducing apoptosis, and upregulating the expression of tumor suppressor genes such as p21cip/WAF [14]. Most parasites have complex life cycles and exhibit diverse morphologies, suggesting that gene expression is epigenetically regulated. Additionally, parasites‘ own HDACs are implicated in their survival and reproduction [15]. Therefore, HDAC is regarded as a potential target for anti-parasitic therapy.

In vitro studies have revealed that the hydroxamic acid HDACi vorinostat is a pan-histone deacetylase inhibitor capable of inhibiting the enzyme activity of HDAC1-3 and HDAC6 [16]. Vorinostat has exhibited certain therapeutic efficacy against diverse cancers and was approved in the United States in 2006 for the treatment of cutaneous T cell lymphoma [17,18]. It has shown potential as a prospective anti-protozoa agent. Vorinostat has previously been demonstrated to inhibit the in vitro growth of *Babesia gibsoni*, *Plasmodium knowlesi*, and *Leishmania donovani*, and it is active against *Cryptosporidium tyzzeri* and *P. falciparum* in vivo [19,20,21,22]. A previous study indicated that vorinostat affected the endodyogeny process and impaired the budding of daughter cells of *T. gondii* [23]. However, whether it possesses anti-*Toxoplasma* activity in vivo and the mechanism underlying its anti-*Toxoplasma* effect remain largely unknown. In this study, we revealed that vorinostat significantly compromised the invasion, replication, and egress of *T. gondii*, thereby severely impairing the lytic cycle. Furthermore, in a murine model, vorinostat showed potential efficacy against early-stage *Toxoplasma* infection. Our data suggested that vorinostat might be a candidate for the treatment of toxoplasmosis.

## 2. Results

### 2.1. Vorinostat Inhibited T. gondii Growth In Vitro

We first analyzed the toxicity of vorinostat on HFF cells. When host cells were exposed to 1 µM vorinostat, the cell survival rate reached 97.4%; when the cells were exposed to 10 µM vorinostat, the survival rate dropped to 83.69% (Figure 1A). The half maximal cytotoxic concentration (CC_50_) of vorinostat against HFF cells was determined to be 214 µM (Figure 1A). Hence, in this study, the safe concentration of vorinostat for HFF cells was less than 1 µM.

To assess the potential of vorinostat against *T. gondii* in vitro, we analyzed the virulence of tachyzoites of the RH strain pre-incubated with vorinostat (1 µM). As shown in Figure 1B, vorinostat effectively compromised the activity of extracellular parasites in a time-dependent manner. Next, we detected the fluorescence intensity of RH-GFP on host cells following treatments with vorinostat at various concentrations. Vorinostat was capable of inhibiting intracellular parasite growth, with an IC_50_ value of 260.1 nM, yielding a selectivity index (=CC_50_/IC_50_) > 800 (Figure 1C). Additionally, the indirect immunofluorescence assay (IFA) also showed a significant decrease in the fluorescence intensities of parasites treated with vorinostat (1 μM) compared to the vehicle group (*p* < 0.05) (Figure 1D). Moreover, the inhibitory effect of vorinostat was more obvious than that of pyrimethamine (2.5 μM) (Figure 1D). These results indicate that vorinostat inhibits activities of both extracellular and intracellular *T. gondii* tachyzoites.

### 2.2. Vorinostat Impaired the Lytic Cycle of T. gondii In Vitro

*The* virulence of *T. gondii* depends on its invasion and intracellular replication. To explore the effects of vorinostat on the entire process of the *T. gondii* lytic cycle, we performed invasion, cell cycle, replication, and plaque assays on HFF cells. For the invasion assay, vorinostat significantly reduced the invasion rate of RH tachyzoites (Figure 2A). Subsequently, by employing flow cytometry, the distributions of cell cycle phases in parasites were determined by measuring DNA content in vorinostat-treated or vehicle-treated parasite populations. As shown in Figure 2B, we found that the percentage of parasites in the G1 phase decreased from 80.91 ± 3.12% in vorinostat-treated parasites to 61.29 ± 6.80% in the vehicle-treated parasite population after 24 h of infection. In contrast, the percentage of parasites in the S phase increased from 12.35 ± 2.97% in vorinostat-treated parasites to 32.92 ± 4.29% in vehicle-treated parasites. The percentage of parasites in the M phase also decreased from 6.75 ± 1.02% vorinostat-treated parasites to 5.79 ± 2.79% in vehicle-treated parasites. These data indicated that vorinostat induced G1 and M arrest.

To assess the inhibitory effects of vorinostat on intracellular proliferation in vitro, the quantity of individual parasites per vacuole was examined. Figure 2C indicated that vorinostat significantly impaired intracellular parasite replication, with a marked decrease in the number of vacuoles containing eight or more parasites. Additionally, we monitored parasite growth through plaque assays in the presence or absence of 1 µM vorinostat. As shown in Figure 2D, almost no obvious plaques were found in the vorinostat-treated group. Taken together, these findings suggested that vorinostat effectively inhibited the lytic cycle of *T. gondii* in vitro.

### 2.3. Vorinostat Profoundly Altered Protein Expression of T. gondii

To further investigate the impact of vorinostat on *T. gondii* protein expression, a data-independent acquisition (DIA)-based quantitative proteomics analysis was conducted. *T. gondii* was incubated with 1 μM vorinostat or 0.1% DMSO (vehicle control) for 12 h. As shown in Figure 3A, over 5000 proteins were identified in each sample. Compared to the vehicle group, a total of 111 proteins were significantly upregulated, while 34 were significantly downregulated (Figure 3B and Appendix A). The heatmap generated based on differentially expressed proteins clearly distinguished the vorinostat-treated group from the vehicle group (Figure 3C). Subsequently, we performed a GO enrichment analysis of differentially expressed proteins. As illustrated in Figure 3D, the top 10 most enriched GO terms are listed, with “DNA replication” (GO:0006260), “membrane” (GO:0016020), and “DNA binding” (GO:0003677) being among the top three most prominent GO terms. KEGG pathway analysis revealed 10 significantly enriched pathways, with differential expressed proteins primarily involved in “DNA replication”, “Toxoplasmosis”, and “Autophagy” pathways (Figure 3E).

### 2.4. Vorinostat Induced Reactive Oxygen Species (ROS) Generation and Promoted Apoptosis in T. gondii

Exposure to HDACis has been reported to arrest cell growth, increase DNA damage, and activate apoptotic pathways. ROS, key markers of oxidative stress, have been established as a critical mechanism in HDACi-induced apoptosis. In this study, we analyzed ROS production in *T. gondii* treated with vorinostat for 24 h using the Oxidant-sensing Probe DCFH-DA. Flow cytometry results demonstrated that vorinostat significantly induced ROS generation in *T. gondii* (*p* < 0.0001) (Figure 4A). Additionally, Annexin V-FITC and PI double staining were employed to assess apoptosis. As shown in Figure 4B,C, treatment with vorinostat significantly increased early apoptosis (Annexin V+/PI−) in parasites compared to the vehicle group (*p* < 0.05).

### 2.5. Anti-Toxoplasma Activity of Vorinostat In Vivo

To evaluate the in vivo effects of vorinostat on *T. gondii*, C57BL/6 mice were intraperitoneally infected with either 100 *T. gondii* RH strain tachyzoites or 500 ME49 strain tachyzoites. After 12 or 24 h post-infection, mice received a daily dose of 25 mg/kg of vorinostat or vehicle for seven consecutive days. Animals in the mock group were intraperitoneally injected with sterile PBS instead of tachyzoites and treated with vorinostat as described above. All mice in the mock group survived throughout the experiment. In the vehicle-treated group, an infection dose of 100 RH tachyzoites resulted in 100% mortality by day 9 post-infection (Figure 5A). However, treatment with 25 mg/kg vorinostat prolonged survival time. Specifically, administration of vorinostat at 12 h post-infection resulted in an 80% survival rate among mice infected with 500 ME49 parasites (Figure 5B). Although vorinostat treatment initiated at 24 h post-infection did not prevent mortality, it slowed the disease progression and extended survival time.

## 3. Discussion

In *T. gondii*, five class I and II HDAC homologous proteins and two SIR2 subtype homologues (class III HDAC) have been identified [24,25]. However, the biological functions of most of these proteins have not been fully elucidated. They might play a role in regulating gene transcription, like those well-known and extensively investigated human HDACs. Research targeting epigenetic regulatory enzymes for the prevention and treatment of parasitic diseases has garnered growing attention in recent years. Here, we discovered that vorinostat significantly mitigated the pathogenesis of *T. gondii* by inhibiting invasion, replication, and egress, thereby severely disrupting the lytic cycle. Notably, in a murine model, vorinostat proved effective for early-stage *Toxoplasma* infection.

Drug repurposing approaches have successfully employed HDACis as cancer therapeutics and are currently being extensively investigated for their potential in treating various *Toxoplasma* infections [26]. To date, several HDACis have demonstrated anti-*Toxoplasma* efficacy with low 50% inhibitory concentration (IC_50_) values and high selectivity indexes. FR235222, along with its derivatives W363 and W399, acts directly on TgHDAC3, thereby inhibiting tachyzoite growth in vitro [27]. In recent years, an increasing number of hydroxamic acid-based HDACis have been discovered through rational design and have demonstrated potent anti-*Toxoplasma* activity in vitro and in vivo. Zhang et al. disclosed that panobinostat impeded the proliferation of RH tachyzoites in a dose-dependent manner, with a selective index of 19.62. After being treated with panobinostat, the symptoms of ocular toxoplasmosis were obviously relieved in a mouse model [28]. Furthermore, MC1742, MC2590, and MC22125 had an IC_50_ lower than 100 nM and a selectivity index of more than 100. In vivo, these three compounds were effective in preventing the outcome of acute toxoplasmosis in a mouse model [29,30]. Thus, HDACis exhibit significant potential in the fight against toxoplasmosis. It is essential to screen for more HDACis that demonstrate high efficacy while maintaining low toxicity levels.

In this study, vorinostat not only had an inhibitory effect on extracellular parasites but also compromised the activity of intracellular parasites. These data suggest that vorinostat might be directly incorporated into the parasitophorous vacuole (PV), where *T. gondii* resides and replicates. We utilized a luciferase-expressing parasite and revealed that vorinostat exhibited an IC_50_ value of 260.1 nM against *T. gondii*, which is consistent with a previous report [30]. The *T. gondii* cell cycle consists of three major phases, namely the G1 phase, S phase, and M phase (mitosis). Cell division time varies among different strains and genotypes. The variations in replication time are primarily influenced by the G1 growth phase, during which precursor nucleotides and enzymes essential for replication are synthesized [31]. Importantly, the regulation of cell mitosis is regarded as a potentially effective strategy for controlling eukaryotic pathogens [32]. In several malignancies, including lymphoma, glioma, and sarcoma, vorinostat had selective anticancer effects, which were attributed to cell cycle arrest in the G1 or G2/M phases [33,34]. In this study, we found that vorinostat exposure resulted in a significant increase in the accumulation of cells at the G1 and M phases, and the distribution of cells at the S phase decreased, which suggest that one of the mechanisms through which vorinostat may exert its inhibitory effects on the proliferation of *T. gondii* is by impeding cell cycle progression. Additionally, proteome analysis found that differentially expressed proteins were significantly enriched in “DNA replication”. These data might indicate DNA damage is accompanied by arrest in the parasite cell cycle. Furthermore, TgHDAC2 (TGRH88_063370) was identified as one of the dysregulated proteins following vorinostat treatment (Appendix A). Previous studies have demonstrated that TgHDAC2 regulates lactylation, acetylation, and 2-hydroxybutyrylation [35]. Whether TgHDAC2 serves as the target enzyme for vorinostat requires further investigation.

ROS encompass a range of highly bioactive oxidative compounds generated either by external oxidizing agents or during cellular aerobic metabolism. Previous studies have demonstrated that HDACi treatment increases ROS production, which has been recognized as a critical mediator of DNA damage and apoptosis [36]. In this study, vorinostat was found to induce intracellular ROS generation in *T. gondii*. Moreover, Annexin V/PI staining results revealed a significant increase in the percentage of apoptotic cells following vorinostat treatment. The balance between ROS production and antioxidant defenses is delicate and can be disrupted by excessive ROS generation. Excess ROS can cause direct damage to lipids, proteins, and pigments, ultimately leading to membrane damage [37]. Differentially expressed proteins induced by vorinostat treatment were significantly enriched in the “membrane” (GO:0016020). A previous study showed that vorinostat incubation led to incomplete IMC assembly and abnormalities in the endodyogeny process, potentially due to excess ROS during treatment [23]. Our findings suggest that vorinostat promotes apoptotic induction through mechanisms involving cell cycle arrest and the impairment of antioxidant defenses, including DNA damage and increased ROS production. However, we fully acknowledge that the precise mechanism by which vorinostat exerts its effects on *T. gondii* remains to be fully elucidated. The intricate interactions between vorinostat, the parasite, and the host cellular environment necessitate further investigation.

In this study, we demonstrated that vorinostat exhibited significant therapeutic efficacy against *T. gondii* infection in vivo. Notably, the early administration of vorinostat protected 80% of mice infected with the low-virulence ME49 strain. However, the study had important limitations. First, vorinostat showed limited efficacy in protecting mice against high-virulence strains. Second, to achieve comparable in vivo activity to sulfadiazine sodium, pyrimethamine, or their combination, further chemical optimization of vorinostat is required. Additionally, combination therapy has long been an important basis for improving the treatment of a variety of complex diseases, including toxoplasmosis. In the future, the combination use of vorinostat with other agents should be evaluated. Third, although no apparent toxic effects were observed in the mock-infected group, the potential adverse effects of vorinostat during treatment should be further evaluated, as it has been associated with gastrointestinal symptoms, fatigue, thrombocytopenia, and thrombosis in clinical trials [38]. Fourth, the stage conversion between rapidly dividing tachyzoites and slowly replicating encysted bradyzoites is a critical event in the pathogenesis of toxoplasmosis. Bradyzoites in tissue cysts are refractory to currently available *Toxoplasma* treatments. Therefore, additional in vivo experiments are necessary to assess the therapeutic potential of vorinostat for chronic toxoplasmosis.

## 4. Materials and Methods

### 4.1. Animals and Parasites

Six- to eight-week-old female C57BL/6 mice weighing 15–20 g were obtained from the Laboratory Animal Center of Shandong University, China, and acclimated to the animal facility for one week. All mice were housed in microisolator cages under specific pathogen-free (SPF) conditions with ad libitum access to food and water.

*T. gondii* RH strain (type I) and ME49 strain (type II) were maintained in confluent monolayers of human foreskin fibroblast (HFF) cells (HS27; ATCC: CRL-1634, Manassas, VA, USA). The cells were cultured in Dulbecco’s Modified Eagle Medium (DMEM) (Cellmax, Beijing, China) supplemented with 10% fetal bovine serum (FBS) (Cellmax, Beijng, China), 2 mM glutamine, 100 U/mL penicillin, and 10 µg/mL streptomycin (Servicebio, Wuhan, China) at 37 °C in a 5% CO_2_ incubator.

### 4.2. Compounds

Vorinostat (CAS: 149647-78-9) was purchased from MedChemExpress (MCE, Shanghai, China). It was dissolved in dimethyl sulfoxide (DMSO, CAS: 67-68-5) (Solarbio, Beijing, China) and stored at −80 °C until use. Pyrimethamine (CAS: 58-14-0) was purchased from Sigma-Aldrich (St. Louis, MO, USA).

### 4.3. In Vitro Cytotoxicity Study

The cytotoxicity of vorinostat on HFF cells was evaluated using the Cell Counting Kit-8 (CCK-8; C0005, TargetMol, Boston, MA, USA) [39]. Briefly, HFF cells were seeded in 96-well plates at a density of 1 × 10^4^ per well and then incubated for 24 h. Cells were then exposed to vorinostat at final concentrations of 0.1, 1, 10, 25, 50, 100, 250, 500, 1000, 2000, and 5000 μM for 48 h. A total of 10 μL CCK-8 reagent was added to each well, followed by an additional 3 h incubation at 37 °C. Finally, the absorbance of the supernatant from each well was measured at 450 nm using an ELx800 microplate reader (BioTek Inc., Winooski, VT, USA).

### 4.4. Parasite Quantification

Parasite proliferation was quantified using an absolute quantitative PCR method as previously described [40]. Briefly, genomic DNA was extracted from the samples and amplified with specific primers targeting the *T. gondii* B1 gene (5′-TGAGTATCTGTGCAACTTTGG-3′ and 5′-TCTCTGTGTACCTCTTCTCG-3′). A standard curve was generated using 10-fold serial dilutions of gDNA extracted from 10^7^ parasites. Parasite burden in each sample was determined relative to this standard curve using a StepOne real-time PCR machine with FastKing One Step RT-PCR MasterMix (Tiangen, Beijing, China).

### 4.5. In Vitro Evaluation of Vorinostat on T. gondii Activity

Freshly purified RH tachyzoites were pre-incubated with vorinostat (1 μM) or 0.1% DMSO (vehicle control) for 1, 6, and 12 h, respectively. Approximately 200 parasites from each treatment group were inoculated onto confluent HFF monolayers in 12-well plates for 48 h. Parasite replication was assessed by an absolute quantitative PCR method as described above. Subsequently, we determined the IC_50_ of vorinostat against *T. gondii*. Freshly purified tachyzoites of *T. gondii* RH-GFP strain (MOI 5) were seeded onto HFF cell monolayers for 3 h. The coverslips were then washed three times with PBS to remove extracellular parasites, and the cells were treated with vorinostat at final concentrations of 1, 50, 100, 200, 300, 400, 500, and 1000 nM for 24 h. Imaging was conducted on a Zeiss Axio Vert.A1 microscope (Carl Zeiss AG, Oberkochen, Germany) equipped with a 20× objective lens, and images were captured using ZEN imaging software (software 2.1, Carl Zeiss, Germany).

### 4.6. In Vitro Invasion Assay

Freshly purified RH tachyzoites (MOI 1) were seeded onto monolayers of HFF cells for 3 h in medium containing vorinostat (1 μM) or DMSO (as the vehicle). Subsequently, the coverslips were washed with PBS to remove extracellular parasites, and each well was fixed with 4% paraformaldehyde (BL539A, Biosharp, Shanghai, China) for 20 min. Mouse anti-SAG2 polyclonal antibody (prepared in our laboratory) diluted 1:500 and Alexa Fluor 594-conjugated goat anti-mouse IgG (A0521, Beyotime, Shanghai, China) were then used to count the number of attached parasites. After permeabilization with 0.1% Triton X-100 (CAS: 9002-93-1, Solarbio, Beijing, China), mouse anti-IMC1 polyclonal antibody (obtained by our laboratory) diluted 1:500 and Alexa Fluor 488-conjugated goat anti-mouse IgG (A-11001, Invitrogen, Waltham, MA, USA) were utilized to count the number of invading parasites. Cell nuclei were stained with 4′,6-Diamidino-2-phenylindol (DAPI, CAS: 28718-90-3, Solarbio, Beijing, China). The numbers of host nuclei, extracellular parasites, and the whole parasites were counted in at least 6 randomly selected visual fields from each slide. Invasion rate = (number of the whole parasite-number of extracellular parasites)/number of host nuclei.

### 4.7. In Vitro Replication Assay

Freshly purified RH tachyzoites (MOI 1), which were pretreated with medium containing vorinostat (1 μM) or 0.1% DMSO (as the vehicle) for 24 h, were seeded onto monolayers of HFF cells for 24 h. Tachyzoites in parasitophorous vacuoles were marked with mouse anti-IMC1 as described above and counted in at least 100 vacuoles per slide.

### 4.8. Plaque Assay

Approximately 200 parasites were inoculated onto confluent monolayers of HFF cells in 12-well plates (Thermo Fisher Scientific, Shanghai, China) and grown for 8 days in medium containing vorinostat (1 μM) or not (vehicle). The host cells were fixated with 4% paraformaldehyde (BL539A, Biosharp, Beijing, Chian) and then stained with the 0.1% Crystal Violet solution (G1063, Solarbio, Beijing, China). The plates were scanned to calculate the number and average area of the plaques using Image J software (https://imagej.net/ij/ (accessed on 10 January 2024)).

### 4.9. Cycle Analysis

The Cell Cycle and Apoptosis Analysis Kit (C1052, Beyotime, Shanghai, China) was utilized to assess the cell cycle. Freshly egressed RH tachyzoites were inoculated onto monolayers of HFF cells in medium containing vorinostat (1 μM) or 0.1% DMSO for 24 h. The host cells were manually disrupted by syringe lysis through a 27G needle. Tachyzoites were then collected, centrifuged, washed, and resuspended in ice-cold 70% ethanol for fixation and subsequently stained with propidium iodide (PI) [41]. The cell cycle was monitored on a Cytoflex flow cytometer (Beckman Coulter, Brea, CA) with 10,000 events per determination and analyzed with ModFit LT software version 4.0 (Verity, Topsham, ME, USA).

### 4.10. Quantitative Proteome Analysis

Freshly purified tachyzoites of RH strain were harvested, centrifuged, washed, and treated with vorinostat (1 μM) or 0.1% DMSO (vehicle) for 12 h. After sample collection, proteome sequencing was conducted by Novogene Co., Ltd. (Beijing, China) as previously described [42]. Briefly, samples were lysed with DB lysis buffer (8 M Urea, 100 mM TEAB, pH 8.5). The supernatant of the lysate was supplemented with 1 M DTT for a 1 h reaction at 56 °C and subsequently alkylated. Then, all of samples were digested with trypsin following quality assessment. The sample eluents were collected and lyophilized. Next, LC-MS/MS analysis was executed using a Vanquish Neo upgraded UHPLC system and a Thermo orbitrap astral mass spectrometer. The raw files were searched and analyzed using the DIA-NN library search software in accordance with the *T. gondii* reference protein database (https://toxodb.org/toxo/app) (accessed on 10 January 2024). Protein quantification results were statistically analyzed using Student’s *t*-test. Proteins with *p* values <  0.05 and FC  >  1.2 or FC  <  0.83 were considered significantly altered. GO enrichment analysis was performed by employing web-based GO software (https://geneontology.org/) (accessed on 14 January 2024) [43]. Pathway enrichment analysis was carried out by using the web-based Kyoto Encyclopedia of Genes and Genomes database (KEGG, https://www.genome.jp/kegg/) (accessed on 14 January 2024) [44].

### 4.11. Detection of Reactive Oxygen Species (ROS)

The ROS levels were measured by employing the Reactive Oxygen Species Assay Kit (S0033S, Beyotime, Shanghai, China) as previously described [45]. Briefly, freshly purified RH tachyzoites were treated with vorinostat (1 μM) or DMSO for 24 h. Parasites were collected and stained with 10 μM of DCFH-DA at 37 °C for 20 min. The fraction of DCF fluorescence was detected using the Cytoflex flow cytometer (Beckman Coulter, Boulevard Brea, CA, USA).

### 4.12. Apoptosis Analysis

The apoptosis of RH tachyzoites was determined with the Annexin V-FITC Apoptosis Detection Kit (C1062S, Beyotime, Shanghai, China) as previously described [46]. Fresh purified parasites were treated with vorinostat (1 μM) or 0.1% DMSO (vehicle) for 24 h. Parasites were collected and then resuspended in 195 uL binding buffer, followed by incubation with 5 μL Annexin V-FITC and 10 μL PI at room temperature in the dark for 15 min. The Cytoflex flow cytometer (Beckman Coulter) was utilized to analyze labeled parasites. Annexin V-FITC-positive and PI-negative (Annexin V-FITC+/PI−) cells are early apoptotic cells, and Annexin V/PI-double-positive are identified as late apoptotic cells.

### 4.13. Anti-Toxoplasma Activity In Vivo

Female C57BL/6 mice were intraperitoneally injected with 100 RH tachyzoites or 500 tachyzoites of the ME49 strain. After 12 or 24 h, the animals were intraperitoneally injected with vorinostat (25 mg/kg in 10% DMSO, 40% PEG-300, 5% Tween 80, and saline) or vehicle (10% DMSO, 40% PEG-300, 5% Tween 80, and saline) for 7 days. Meanwhile, the animals in the mock group were intraperitoneally injected with the same volume of sterile PBS without tachyzoites and then treated with vorinostat (25 mg/kg in 10% DMSO, 40% PEG-300, 5% Tween 80, and saline, i.p.) for 7 days. Mice were euthanized upon reaching humane endpoints.

### 4.14. Statistical Analysis

All statistical analyses in this study were performed using GraphPad Prism 7.0 software (GraphPad Software, San Diego, CA, USA). Results were presented as mean ± standard deviation (SD). Significance was determined by Student’s *t*-test (two-tailed, unpaired), one-way ANOVA, or two-way ANOVA. A *p* value < 0.05 indicated statistical significance.

## 5. Conclusions

This study disclosed that vorinostat efficaciously inhibited the lytic cycle of *T. gondii*, arrested the parasite cell cycle in the G1 and M phases, and elicited parasite apoptosis, thereby resulting in the suppression of parasite proliferation. The study also demonstrated that vorinostat exhibited a significant in vivo anti-*Toxoplasma* effect, indicating its potential as a candidate for the development of novel therapeutic strategies against toxoplasmosis.

## Figures and Tables

**Figure 1 ijms-26-00795-f001:**
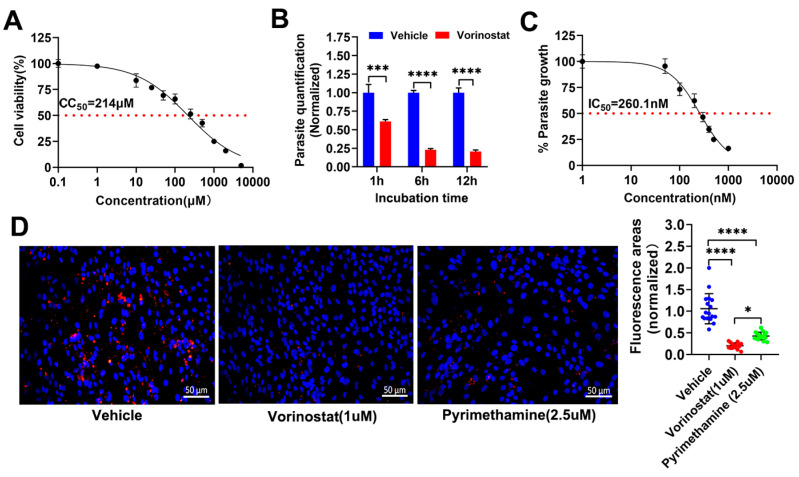
Vorinostat inhibits the replication of *T. gondii* tachyzoite in vitro. (**A**) Assessment of HFF cell viability upon treatment with vorinostat. CC_50_ was examined; (**B**) Effects of vorinostat on the activity of extracellular RH tachyzoites of *T. gondii*. Freshly purified RH tachyzoites were pretreated with vorinostat (1 μM) or vehicle (0.1% DMSO) for 1, 6, or 12 h and subsequently inoculated onto monolayers of HFF cells (MOI 1) and cultured for 48 h. Samples were collected for quantification. The *p* values by two-way ANOVA are indicated; *** *p* < 0.001 and **** *p* < 0.0001; (**C**) Inhibition of RH-GFP strain tachyzoites following treatment with vorinostat. The IC_50_ was determined. (**D**) Inhibition of RH strain tachyzoites (MOI 1) after treatment with vorinostat (1 μM). HFF cells infected with RH tachyzoites were treated with vorinostat (1 μM), pyrimethamine (2.5 μM), or 0.1% DMSO (vehicle) for 24 h. The infected HFF cells were stained with DAPI and rat anti-SAG2 antibody. Representative immunofluorescence images are shown. The red fluorescence areas of per image were used as a metric of parasite intracellular growth. Normalized values represent means ± SD, 5 fields from each of 3 independent experiments (n = 15). The *p* values by one-way ANOVA are indicated; * *p* < 0.05 and **** *p* < 0.0001.

**Figure 2 ijms-26-00795-f002:**
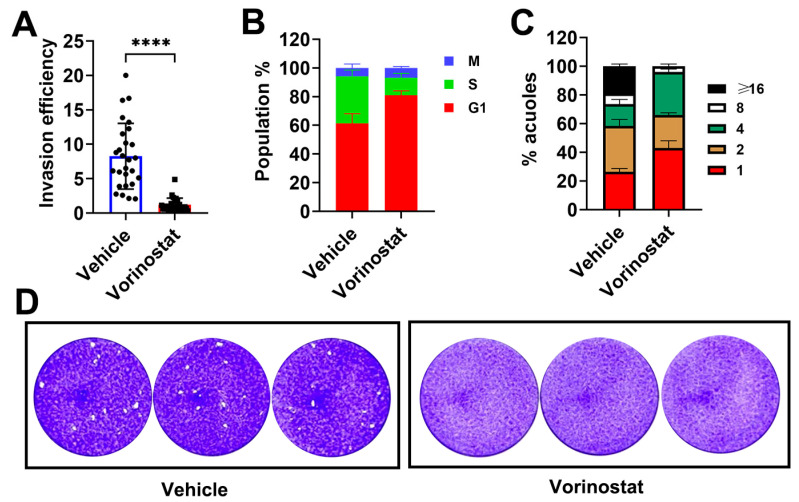
Vorinostat demonstrates inhibitory effects at multiple stages of the *T. gondii* lytic cycle. (**A**) Invasion assay. Invasion events were determined using two-color staining to differentiate invaded from non-invaded parasites. Data represent the means ± SD from three independent experiments conducted in duplicate. *p* values by unpaired two-tailed Student’s *t*-test are indicated; **** *p* < 0.0001. (**B**) Cell cycle analysis. RH tachyzoites were exposed to vorinostat (1 µM) for 24 h, and changes in the cell cycle were analyzed via flow cytometry. Results represent means ± SD from three independent experiments performed in triplicate. (**C**) Replication assay. RH strain tachyzoites were incubated with either 0.1% DMSO (vehicle control) or vorinostat (1 µM). The number of parasites within 100 randomly selected vacuoles was counted. Data are presented as the means ±SD from at least three independent experiments. (**D**) Plaque formation assay. RH strain tachyzoites were treated with or without vorinostat (1 µM) for 8 days.

**Figure 3 ijms-26-00795-f003:**
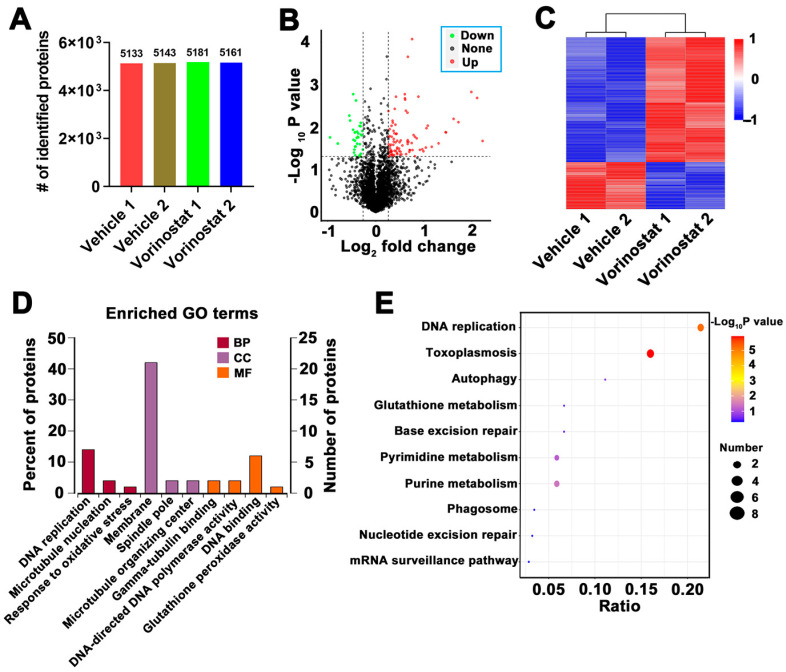
Proteome analysis of the inhibitory effect of vorinostat on RH tachyzoites. (**A**) Number of identified proteins in each sample. (**B**) Volcano plots of differentially expressed proteins. Red and green dots represent significantly up- and downregulated proteins, respectively. The X-axis indicates log2-fold change, while the Y-axis shows −log10 (*p* value). (**C**) Heatmap constructed based on differentially expressed proteins. Columns are hierarchically clustered using complete linkage with Pearson correlation coefficient as the distance measure. (**D**) Gene Ontology (GO) analysis. GO terms are categorized into three aspects, namely molecular function (MF), biological process (BP), or cellular component (CC). The X-axis label represents the corresponding GO terms, while the left Y-axis label denotes the percentage of proteins involved. Additionally, the right Y-axis label indicates the number of proteins. (**E**) KEGG pathway analysis. The top 10 significantly enriched KEGG pathways are listed. The color intensity of the points represented the *p* value from the hypergeometric test, with a gradient from blue to red indicating increasing statistical significance. The size of each dot reflects the number of differentially expressed proteins in the corresponding pathway, and the larger dots indicate a higher number of proteins involved. The X-axis label shows the rich ratio, and the Y-axis label displays the KEGG pathway terms. The color of the dots represents −log10 (*p* value), and the size of the dot reflects the protein number.

**Figure 4 ijms-26-00795-f004:**
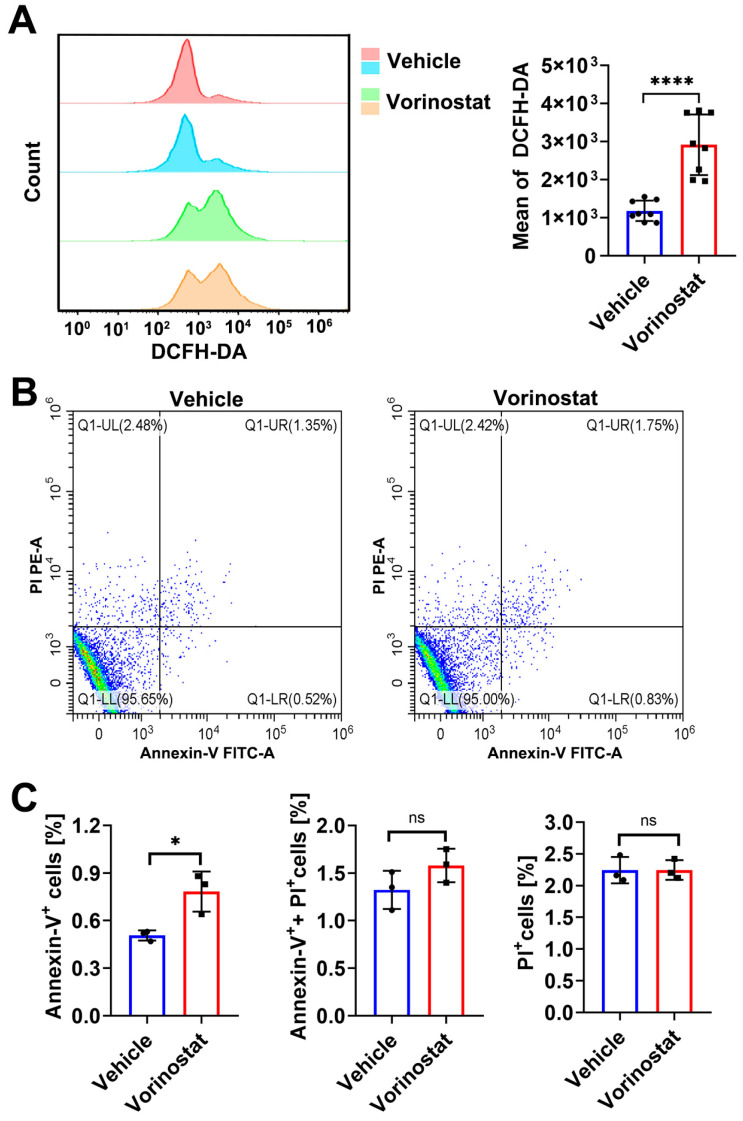
Vorinostat promotes ROS production and induces apoptosis in *T. gondii*. (**A**) Assessment of ROS production. Tachyzoites of *T. gondii* RH strain were incubated with vorinostat or 0.1% DMSO (vehicle) for 24 h. Flow cytometry analysis was then performed using DCFH-DA staining to determine intracellular ROS levels. Data are presented as the means ± SD from at least three independent experiments. Statistical significance was determined by Student’s *t*-test; **** *p* < 0.0001. (**B**) Representative flow cytometric dot plots of Annexin V-FITC/PI staining are shown. (**C**) The percentage of apoptotic parasites following treatment with vorinostat or vehicle was quantified via flow cytometry. Mean percentages ± SD of Annexin V+, PI+, or PI− parasites were calculated from least three independent experiments. Statistical significance was determined by Student’s *t*-test; * *p* < 0.05, ns means not significant.

**Figure 5 ijms-26-00795-f005:**
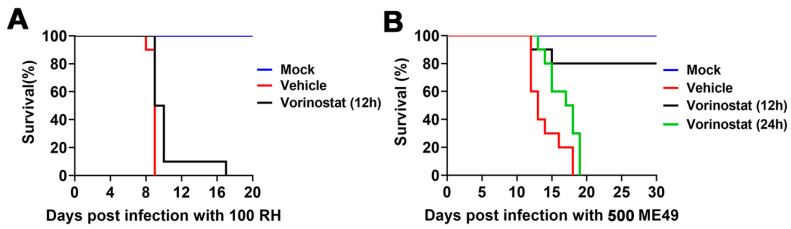
Efficacy of vorinostat against acute murine toxoplasmosis. (**A**) Survival study with RH tachyzoites. Female C57BL/6 mice (n = 10) were intraperitoneally infected with 100 RH strain tachyzoites and subsequently treated with 25 mg/kg vorinostat (black line) or vehicle (red line) once daily for 7 days starting 12 h post-infection. Mice in the mock group (blue line) were intraperitoneally injected with sterile PBS instead of tachyzoites. (**B**) Survival study with ME49 tachyzoites. Female C57BL/6 mice (n = 10) were intraperitoneally infected with 500 ME49 strain tachyzoites and treated with 25 mg/kg vorinostat or vehicle (red line) once daily for 7 days, commencing either 12 h (black line) or 24 h (green line) post-infection. Mice in the mock group (blue line) received intraperitoneal injections of sterile PBS instead of tachyzoites. The survival of mice was monitored and recorded daily.

## Data Availability

The mass spectrometry proteomics data have been deposited into the ProteomeXchange Consortium (https://proteomecentral.proteomexchange.org) (accessed on 15 January 2025) via the iProX partner repository with the dataset identifier PXD053997.

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
