# Peer review of "Characterization of the Activities of Vorinostat Against Toxoplasma gondii"

_ijms, 2025, doi:10.3390/ijms26020795_

Round 1
Reviewer 1 Report
Comments and Suggestions for Authors
In this manuscript the authors report the effects of Vorinostat, a well known inhibitor of HDAC, on Toxoplasma gondii infection in vitro and in vivo. HDAC inhibitors were tested previously in experiments in vitro. The present study reported by Zeng et al. show a significant effect of Vorinostat both in vitro and in vivo, with a prolonged survival of infected mice and with better results than those obtained with pyrimetamine, used as a positive control. It is important to point out that two parasite strains, RH and ME49, were used. The results obtained are of interest and merit to be published, However, there are two points that should be adressed by the authors:
1. It is important to add some images of the infected cells (control and treated);
2. It is important to comment on results obtained with the ME49, especially in relation to the evolution of the number of cysts.
Author Response
Comments 1: It is important to add some images of the infected cells (control and treated).
Response 1: In the Figure 1D, the intracellular parasites were labelled with the rat anti-SAG2 antibody(red). Parasites and host nucleus were visualized by DAPI (blue). In the vehicle group (without drug), there are more parasites in the parasitophorous vacuoles compared with the drug-treated group. In the Figure 2D, Plaque assay results showed that parasites in the vehicle group could grow normally, while plagues were rarely observed when the parasites were treated with vorinostat. The above data indicated that vorinostat could impair the parasites’ virulence and their multiplication capacity in infected host cells.
Comments 2: It is important to comment on results obtained with the ME49, especially in relation to the evolution of the number of cysts.
Response 2: Thanks for your comments. The administration of 25 mg/kg of vorinostat from 12h resulted in 80% survival rate of mice infected with 500 ME49 parasites. At 30 days post infection, we in fact found tissue cysts in the drug-treated mouse brain, which indicated vorinostat can not eradicated parasites from the host. The mice in the vehicle group all died within 18 days post infection when we did not find tissue cysts in mouse brains.
Reviewer 2 Report
Comments and Suggestions for Authors
This study aimed to characterize the activity of vorinostat against T. gondii and evaluate its potential as a new treatment option for toxoplasmosis. This study demonstrated that vorinostat effectively inhibited T. gondii growth both in vitro and in vivo. This finding is significant because the current treatment options for toxoplasmosis are limited and can have toxic side effects. Vorinostat could potentially serve as a new therapeutic option, especially for cancer patients infected with T. gondii. Additionally, vorinostat has been approved for treating certain types of cancer. Its effectiveness against T. gondii suggests that it could be particularly beneficial for cancer patients who are also at a risk of toxoplasmosis. This study provides insights into the effects of vorinostat on T. gondii infection. It impairs the lytic cycle of the parasite, induces cell cycle arrest, increases ROS production, and promotes apoptosis. This understanding of the mechanism of action could lead to further research and the development of targeted therapies. The research indicated that Vorinostat has a high selectivity index, suggesting that it may be effective against T. gondii at concentrations that are not toxic to host cells. This is an important consideration for the development of potential therapeutic agents.
However, further research is required to fully establish the efficacy and safety of vorinostat in humans. The manuscript is acceptable with minor revisions. The following are the comments.
Line 403: There is a lack of in vivo dose-response data. Please clarify the choice of a single dose of vorinostat (25/mg/kg). Without dose-response data, it is difficult to determine the optimal therapeutic dose or potential toxicity at higher doses.
-Proteome analysis identified changes in protein expression, but it is not clear whether these changes are directly caused by vorinostat or are secondary effects of parasite stress or death. Could this lead to misinterpretation of the drug's primary targets? Can you add a comment on this in the Discussion section?
- Limited discussion on potential side effects: While the study mentions vorinostat's selectivity index, there is limited discussion on potential side effects, particularly in the context of its use as an anticancer drug. This information is crucial to assess its potential as a treatment for toxoplasmosis.
- Generalizability of results: This study primarily used the RH strain of T. gondii, with limited testing of the ME49 strain. Given the genetic diversity of T. gondii strains, please clarify the generalizability of results to other strains.
There were some limitations to this study. Suggestions:
1. This study primarily focused on acute infections, with limited exploration of chronic toxoplasmosis.
2. The in vivo studies were relatively short term.
3. The exact mechanism of action is not fully elucidated.
4. Limited comparison with current treatments: Although there is a brief comparison with pyrimethamine, there is no comprehensive comparison with current standard treatments for toxoplasmosis.
Despite these limitations, the premises and findings generally support the conclusions of this study. The authors acknowledge the need for further research, which is appropriate, given the scope of this study.
Minor comments:
Line 16: The use of plural "values" here is somewhat inconsistent with the single value reported in the abstract
Line 218: What was the rationale for choosing different infection doses for the RH (100 tachyzoites) and ME49 (500 tachyzoites) strains?
Line 348: T. gondii should be italicized.
Author Response
Comments 1:Line 403: There is a lack of in vivo dose-response data. Please clarify the choice of a single dose of vorinostat (25/mg/kg). Without dose-response data, it is difficult to determine the optimal therapeutic dose or potential toxicity at higher doses.
Response 1: Thanks for your comments. We agree that in vivo dose-response data is very important for the estimation of the effect and safety of candidate drugs. Vorinostat is a potent pan-HDAC inhibitor which has been widely studied in many mouse models. In previous studies, such as Butler et al Cancer research 60.18 (2000): 5165-5170, Loh et al Br J Pharmacol. 2019 Oct;176(19):3775-3790, and Ma et al Clin Cancer Res. 2010 Nov 1;16(21):5165-76, mice were treated with 25 mg/kg of vorinostat, and drug-related developmental toxicity was not observed.
In the study performed by L. David Wise et al.(doi: 10.1002/bdrb.20104. PMID: 17294457), there were dose-related decreases in mean body weight gains in the 20- and 50-mg/kg/day groups , but no effects on mean food consumption. Treatment-related findings in the 50-mg/kg/day group were limited to a moderate leukopenia and decreased serum ALP, while in the 20-mg/kg/day group only a slight leukopenia was observed. In our study, all mock-infected mice treated with vorinostat (25/mg/kg) showed no signs of clinical disease or weight loss (data not shown) and no mortality throughout the study, which indicate administration i.p. with vorinostat (25/mg/kg) do not induce toxic effect in vivo.
Comments 2:Proteome analysis identified changes in protein expression, but it is not clear whether these changes are directly caused by vorinostat or are secondary effects of parasite stress or death. Could this lead to misinterpretation of the drug's primary targets? Can you add a comment on this in the Discussion section?
Response 2: Thanks for your comment and suggestion. In the Figure 1B, vorinostat effectively impaired the activity of extracellular parasites in a time-dependent manner. We can deduce that differentially expressed proteins identified by proteome analysis are directly caused by vorinostat treatment rather than secondary effects of parasite stress or death. Although vorinostat is a potent pan-HDAC inhibitor in mammalian cells, its targets in T.gondii have not been identified, which need further exploration. The discussion section has been modified according to your suggestion.
Comments 3:Limited discussion on potential side effects: While the study mentions vorinostat's selectivity index, there is limited discussion on potential side effects, particularly in the context of its use as an anticancer drug. This information is crucial to assess its potential as a treatment for toxoplasmosis.
Response 3: Thanks for your comments. The discussion part has been modified.
Comments 4:Generalizability of results: This study primarily used the RH strain of T. gondii, with limited testing of the ME49 strain. Given the genetic diversity of T. gondii strains, please clarify the generalizability of results to other strains.
Response 4: Thanks for your comments. RH strain is a virulent strain, and ME49 strain is a low virulent strain. They are widely used in the studies on activities of anti-Toxoplasma drugs. Here, we found that vorinostat showed anti-Toxoplasma activity in vivo, especially for the early infection. So far, more than 138 unique genotypes have been identified from distinct areas, which show a low genetic diversity. Vorinostat might be active against other genotypes, which need further investigation.
Comments 5:There were some limitations to this study. Suggestions:
- This study primarily focused on acute infections, with limited exploration of chronic toxoplasmosis.
- The in vivo studies were relatively short term.
- The exact mechanism of action is not fully elucidated.
- Limited comparison with current treatments: Although there is a brief comparison with pyrimethamine, there is no comprehensive comparison with current standard treatments for toxoplasmosis.
Despite these limitations, the premises and findings generally support the conclusions of this study. The authors acknowledge the need for further research, which is appropriate, given the scope of this study.
Response 5: Thanks for your comments and suggestions.
Comments 6:Minor comments:
Line 16: The use of plural "values" here is somewhat inconsistent with the single value reported in the abstract
Response 6: Corrected accordingly.
Line 218: What was the rationale for choosing different infection doses for the RH (100 tachyzoites) and ME49 (500 tachyzoites) strains?
Response 6: Before formal experiments, we found that low infection dose sometimes can not cause acute toxoplasmosis, as the innate immunity might help kill the parasites. Additionally, high doses will cause severe acute toxoplasmosis, which significantly impair the potency of candidate drugs.
Line 348: T. gondii should be italicized.
Response 6: Corrected accordingly.
Reviewer 3 Report
Comments and Suggestions for Authors
The manuscript entitled "" authored by presents results on the in vitro and in vivo activity of a selected product on toxoplasma gondii
As positive aspects of the study:
- importance of the topic for the human medical field, the protocol used, the statistical analysis of the results
As negative aspects that should be addressed by the authors:
- the overstatements found in the conclusions (abstract and main text body) regarding the recommendations to treat both cancer and toxoplasmosis in humans
The authors should objectively refer to the scientific value of the results obtained, the study does not refer to any efficacy and safety of any treatment in human oncological pathology, please modify the corresponding text
Please add the study limitations
The authors are recommended to enrich the text of the introductio, the cited references are dated, while more recent literature is available on the Toxoplasma gondii' epidemiology, pathology, associated conditions and also on cancer
Please, revise the content of line 27-57 and upgrade the text in terms of cited literature novelty and also scientific medical vocabulary
Lines 48-57 do not have the scientific vocabulary and content suitable for an ISI journal, cancer therapy overall is not sufficient to justify "an urgent need to develop drugs that can be effective against both T. gondii and tumors at the same time"
lines 58-90 offer a mixture of details on mechanisms, pathologies, compounds, please refine the text and point out what characterizes the present study to underline the novelty.
Discussion section is not sufficient - the authors repeat the results description, while previous studies covering the same topic are not even mentioned
- please correct the statement regarding the safety as this aspect was evaluated exclusively in mice
- please develop the text by adding previous reports comments
- please add the study limitations
Revise the conclusions in an objective manner and remove the cancer treatment aspects
As a minor observation, please use Italic style for "in vitro", "in vivo" for the whole manuscript
The reference list should be upgraded - recent studies for the introduction and discussion as mentioned above
Author Response
Comments 1:the overstatements found in the conclusions (abstract and main text body) regarding the recommendations to treat both cancer and toxoplasmosis in humans.
Response 1: We are very grateful to expert and reviewer Comments s out these constructive comments to us on the positive and negative aspects. For the negative aspects, we have revised and modified our conclusions.
Comments 2:The authors should objectively refer to the scientific value of the results obtained, the study does not refer to any efficacy and safety of any treatment in human oncological pathology, please modify the corresponding text.
Response 2: Modified accordingly.
Comments 3:Please add the study limitations.
Response 3: Modified accordingly.
Comments 4:The authors are recommended to enrich the text of the introductio, the cited references are dated, while more recent literature is available on the Toxoplasma gondii' epidemiology, pathology, associated conditions and also on cancer.
Response 4: Modified accordingly.
Comments 5:Please, revise the content of line 27-57 and upgrade the text in terms of cited literature novelty and also scientific medical vocabulary.
Response 5: Modified accordingly.
Comments 6: Lines 48-57 do not have the scientific vocabulary and content suitable for an ISI journal, cancer therapy overall is not sufficient to justify "an urgent need to develop drugs that can be effective against both T. gondii and tumors at the same time".
Response 6: Thanks for your comments. The corresponding part has been modified.
Comments 7:Lines 58-90 offer a mixture of details on mechanisms, pathologies, compounds, please refine the text and Comments out what characterizes the present study to underline the novelty.
Response 7: Modified accordingly.
Comments 8:Discussion section is not sufficient - the authors repeat the results description, while previous studies covering the same topic are not even mentioned.
Response 8: Thanks for your suggestion. The discussion part has been modified according to your suggestion.
Comments 9: Please correct the statement regarding the safety as this aspect was evaluated exclusively in mice.
Response 9: Corrected accordingly.
Comments 10:Please develop the text by adding previous reports comments.
Response 10: Modified accordingly.
Comments 11:Please add the study limitations
Response 11: The limitations have been added in the revised MS.
Comments 12:Revise the conclusions in an objective manner and remove the cancer treatment aspects
Response 12: Modified accordingly.
Comments 13:As a minor observation, please use Italic style for "in vitro", "in vivo" for the whole manuscript
Response 13: Corrected accordingly.
Comments 14:The reference list should be upgraded - recent studies for the introduction and discussion as mentioned above
Response: Modified accordingly.
Round 2
Reviewer 3 Report
Comments and Suggestions for Authors
The revised form of the manuscript contains little revision
The authors used track changes style to suggest modifications, but in fact these modifications use the same or similar content as the one that is deleted
Some text modifications involving the use of synonyms is not sufficient and it is not what was recommended with the first review
I recommend the authors to perform the following modifications of the manuscript and to specify the lines in their reply form
1.. enrich the text of the introduction, the cited references are dated, while more recent literature is available on the Toxoplasma gondii' epidemiology, pathology, associated conditions and also on cancer.
2. revise the content of line 27-57 and upgrade the text in terms of cited literature novelty and also scientific medical vocabulary
3. the novelty is not presented
4. Discussion section is not sufficient - the authors repeat the results description, while previous studies covering the same topic are not even mentioned. Please develop the text by adding previous reports comments
5. add the study limitations.
6. The reference list should be upgraded - recent studies for the introduction and discussion as mentioned above
Comments on the Quality of English Language
The English should be improved for the whole manuscript
Author Response
Comments 1. Enrich the text of the introduction, the cited references are dated, while more recent literature is available on the Toxoplasma gondii' epidemiology, pathology, associated conditions and also on cancer.
Response 1: Thank you for your constructive suggestion. We have thoroughly revised the Introduction section to incorporate more recent literature on the epidemiology, pathology, associated conditions, and cancer-related aspects of T. gondii. Specifically, we have updated our references to include recent studies such as the global status of T. gondii infection, which highlights the latest trends and prevalence rates worldwide. Additionally, we have included recent insights into the potential role of T. gondii in cancer mechanisms, emphasizing its impact on host cell responses. We have now cited these references in the revised manuscript.
Comments 2. Revise the content of line 27-57 and upgrade the text in terms of cited literature novelty and also scientific medical vocabulary.
Response 2: Thank you for your valuable suggestions. We have revised the content from lines 27 to 57 to incorporate more recent and novel literature, thereby enhancing the comprehensiveness of our study. Furthermore, we have enriched the scientific and medical terminology in lines 28 to 61 to better align with current research standards.
Comments 3. The novelty is not presented.
Response 3: Thanks for your comments. The novelty of this study has been highlighted in the introduction section (lines 91 to 97)
Comments 4. Discussion section is not sufficient - the authors repeat the results description, while previous studies covering the same topic are not even mentioned. Please develop the text by adding previous reports comments.
Response 4: Thank you for your constructive suggestion. The discussion part has been modified. Some previous studies covering the same topic have been added (line 254-257).
Comments 5. add the study limitations.
Response 5: Thank you for your suggestion. We have modified our manuscript and have added the study limitations in the Discussion section(line 272-274, line 291-306).
Comments 6. The reference list should be upgraded - recent studies for the introduction and discussion as mentioned above.
Response 6: We've updated the reference list with recent studies that fit well with our introduction and discussion sections, making sure our work reflects the latest research.
Comments 7. Comments on the Quality of English Language
The English should be improved for the whole manuscript.
Response 7: We have also thoroughly polished the manuscript. We also have corrected other minor mistakes and errors throughout the manuscript. We hope that these revisions adequately address the concerns you raised.
Round 3
Reviewer 3 Report
Comments and Suggestions for Authors
The authors could have marked the modifications, still v2 and v3 are available and comparable
In their reply, the authors indicate performed revisions, while such revisions do not exist or indicate lines that do not present the requested revisions
Still, for one line interval indicated by the authors and found in v3, the authors wrote:
"several HDACis, namely belinostat, panobinostat, vorinostat, and romidepsin, have already been approved for cancer treatment, thereby making them potential candidates for the management of parasitic diseases[30]. "
This is not sufficient for discussion of a scientific manuscript submitted to a Q1 journal; also, note that cancer treatment approved compounds do not necessarily become candidates for the management of parasitic diseases!
Needless to say that "management of parasitic diseases" involves more measures along with therapy
Finally, the conclusion of the manuscript "vorinostat constitutes a novel lead compound for treating the acute phase of toxoplasmo- sis." is an overstatement i
I believe a true scientific paper should acknowledge study limitations, have pertinent discussions and present results novelty
The authors are not willing to cover such aspects, thus this manuscript is not suitable for publication in a Q1 journal
Author Response
Comments 1: The authors could have marked the modifications, still v2 and v3 are available and comparable
In their reply, the authors indicate performed revisions, while such revisions do not exist or indicate lines that do not present the requested revisions
Response 1: We prepared this manuscript in accordance with the template provided by MDPI journals. Line numbers of text in the template may disappear because of different versions of Office. Meanwhile, we uploaded the PDF of the manuscript, in which the line number exist. In the V3, we performed revisions according to your suggestion. We are sorry that these revisions failed to satisfy your criteria. In the new manuscript, we have refined the text, particularly in the discussion section, to better align with your requirements. These modifications were highlighted in different colors.
Comments 2: Still, for one line interval indicated by the authors and found in v3, the authors wrote:
"several HDACis, namely belinostat, panobinostat, vorinostat, and romidepsin, have already been approved for cancer treatment, thereby making them potential candidates for the management of parasitic diseases[30]. "
This is not sufficient for discussion of a scientific manuscript submitted to a Q1 journal; also, note that cancer treatment approved compounds do not necessarily become candidates for the management of parasitic diseases!
Needless to say that "management of parasitic diseases" involves more measures along with therapy
Response 2: Thank you for your comments on our manuscript. We have added many previous studies covering the same topic, which is highlighted in blue (Line 261-277). Additionally, we agree that "management of parasitic diseases" involves more measures along with therapy, which has been deleted in the new manuscript.
Comments 3:Finally, the conclusion of the manuscript "vorinostat constitutes a novel lead compound for treating the acute phase of toxoplasmo- sis." is an overstatement
Response 3: Lead compounds are chemical compounds that show desired biological or pharmacological activity and may initiate the development of a new clinically relevant compound. This term “lead compund” may be somewhat ambiguous and could potentially be regarded as the optimal therapeutic option. And the conclusion part has been modified.
Comments 4: I believe a true scientific paper should acknowledge study limitations, have pertinent discussions and present results novelty
The authors are not willing to cover such aspects, thus this manuscript is not suitable for publication in a Q1 journal
Response 4: We sincerely apologize for the fact that the modifications made to the manuscript did not meet your expectations. However, we greatly appreciate your suggestions, which we find to be both constructive and instructive. In the new manuscript, we have highlighted the novelty in bright green in the introduction part and the study limitations in yellow in the discussion part. Meanwhile, other deficiencies have been modified, which can be found in the submitted tracked changes version. We genuinely hope that these revisions will effectively address the significant concerns you have raised.
Round 4
Reviewer 3 Report
Comments and Suggestions for Authors
The authors performed the requested revisions